# Structural Changes and Very-Low-Frequency Nonlinear Dielectric Response of XLPE Cable Insulation under Thermal Aging

**DOI:** 10.3390/ma16124388

**Published:** 2023-06-14

**Authors:** Can Wang, Xuetong Zhao, Jingqi Qiao, Yongjian Xiao, Jie Zhang, Yuchen Li, Hanzhong Cao, Lijun Yang, Ruijin Liao

**Affiliations:** 1State Key Laboratory of Power Transmission Equipment & System Security and New Technology, Chongqing University, Shapingba District, Chongqing 400044, China; 202011021109t@cqu.edu.cn (C.W.); 202111131187@stu.cqu.edu.cn (J.Q.); 202111021022@stu.cqu.edu.cn (Y.X.); 202211021082t@cqu.edu.cn (Y.L.); 202211131320@stu.cqu.edu.cn (H.C.); yljcqu@cqu.edu.cn (L.Y.); rjliao@cqu.edu.cn (R.L.); 2China Electronics Chip Technology Research Institute, Chongqing 400044, China; 202011021051@stu.cqu.edu.cn

**Keywords:** XLPE cable insulation, thermal aging, very-low-frequency dielectric property, physicochemical property, *U*–*I* hysteresis curve

## Abstract

The structural changes and very-low-frequency (VLF) nonlinear dielectric responses are measured to evaluate the aging state of cross-linked polyethylene (XLPE) in power cables under various thermal aging conditions. For this purpose, the accelerated thermal aging experiments were performed on XLPE insulation materials at 90 °C, 120 °C and 150 °C with different durations of 240 h, 480 h and 720 h, respectively. The Fourier transform infrared spectrum (FTIR) characterization and differential scanning calorimeter (DSC) were tested to analyze the influence of different aging on physicochemical properties of XLPE insulation. Besides, the VLF dielectric spectra show that the permittivity and dielectric loss change significantly in the VLF range from 1 mHz to 0.2 Hz. A voltage–current (*U–I*) hysteresis curve referring to a standard sinusoidal voltage and the response current were introduced to characterize the nonlinear dielectric properties of XLPE insulation caused by thermal aging.

## 1. Introduction

In recent years, polymer materials have become the focus for their application in electrical devices and power systems [1,2,3]. Power cables take on an important part in electrical transmission and urban distribution networks. Nowadays, cross-linked polyethylene (XLPE) remains a crucial material for high-voltage power cables due to its excellent electrical properties, outstanding physicochemical characteristics, easy processing and low cost [4,5,6]. However, XLPE insulation may suffer irreversible degradation and damage under various stresses. Under normal service conditions, the operating temperature of cable insulation is around 90 °C due to the large ampacity. Sometimes the maximum temperature caused by short circuit or overload can reach 150 °C in a short time [7,8]. Therefore, thermal degradation is still one of the most severe factors that causes cable failure during its service life [9]. To address these issues, massive research studies have been reported on physicochemical characteristics to study the morphology changes of XLPE during thermal aging. It was proposed that the lamellas and spherulites could be destroyed, and free volume could be formed in XLPE insulation during oxidation, resulting in an obvious variation in crystallinity and carbonyl group [10,11]. Thermal oxidation and molecular chain breakage of XLPE eventually lead to insulation breakdown. Therefore, it is very important to detect the aging state of power cables in a timely manner for the safe and stable operation of urban power supply.

Frequency domain spectroscopy (FDS), a nondestructive testing method with high robustness, is widely used to evaluate the performance of XLPE cables based on the evolution of dielectric parameters such as dielectric constant, dielectric loss (tan*δ*) and conductivity with thermal aging [12,13]. Especially, the tan*δ* at the very low frequency (typically at 0.1 Hz) was proposed to evaluate the state of XLPE cables according to IEEE 400.2-2013. Many studies showed that the value of tan*δ* at 0.1 Hz was closely associated with the aging level of cable insulation [14,15]. However, when the aging is not severe, the correlation between the both is weak, and sometimes the condition of the cable insulation will be misdiagnosed based on the value of tan*δ* at a single frequency of 0.1 Hz [16]. Moreover, the severe thermal aging leads to molecular chain breakage of XLPE insulation, and the interfacial polarization and orientation polarization within the insulation in the very-low-frequency range may be enhanced. Therefore, the aging state of cable insulation can be effectively evaluated by the very-low-frequency (VLF) dielectric spectrum. The Debye equation and some modified theory were used to analyze the different polarization mechanisms [17]. However, it is quite time-consuming to test the dielectric loss below 0.1 Hz, limiting its wide application in power systems.

Recently, Yang et al. proposed a Hammerstein–Wiener (H-W) model for improving the dielectric test, which can quickly and accurately obtain the VLF dielectric responses of insulating materials [18]. The measuring time of the VLF dielectric spectrum (1 mHz–1 kHz) is about 70% shorter than that of traditional sweep frequency. The measuring equipment based on the H–W model has been successfully applied to evaluate the oil–paper insulation performance of the power transformer [18], which makes it possible to quickly detect the insulation performance for power equipment under the condition of a short-time outage.

In this work, the structural changes of XLPE insulation with various aging states were characterized using Fourier transform infrared spectroscopy (FTIR) and a differential scanning calorimeter (DSC). Furthermore, the VLF dielectric spectra of XLPE insulation samples were tested from 1 mHz to 1 kHz. The *U*–*I* hysteresis curves at 1 kHz and 1 mHz were extracted to analyze the evolution of the XLPE cable insulation. Finally, the VLF nonlinear dielectric responses of the unaged and aged XLPE cable insulation were discussed.

## 2. Samples and Experiments

### 2.1. Samples Preparation and Thermal Aging Experiment

The XLPE slice samples were cut from a 10 kV commercial cable (YJV62, Shanghai Qifan Cable Co., Ltd., Shanghai, China) using a XLPE cable slicer (KY4289, Yangzhou Kaiyuan Machine Co., Ltd., Yangzhou, China). The XLPE samples were sliced into squares with a side length of 80 mm and a thickness of 0.2 mm along the conductor direction, as shown in Figure 1. Then, the XLPE samples were cleaned in deionized water with an ultrasonic washer to remove impurities and contaminants, and finally dried in a vacuum oven at 70 °C for 24 h.

The XLPE samples were subjected to accelerated thermal aging tests according to the IEC 60216 standard [19]. The samples were suspended vertically in the hot air-circulating ovens (TG220B, Chongqing Wuhuan Machine Co., Ltd., Chongqing, China) at the aging temperatures of 90 °C, 120 °C and 150 °C, respectively. Generally, 90 °C is the maximum working temperature of a power cable under normal working conditions, while 120 °C is the temperature allowed to operate for a period of time in the case of thermal overload, and 150 °C is the temperature at which XLPE can be allowed to work for periods under the condition of short circuit or overload [20,21]. As reported by Li, K. and Afia, R.S.A. et al., 240 h, 480 h and 720 h are the common thermal aging durations used to simulate different degradation of XLPE in laboratories [22,23]. Finally, the samples with different aging states were obtained and divided into Unaged as: group A1, A2, A3; group B1, B2, B3 and group C1, C2, C3, as shown in Table 1.

### 2.2. FTIR and DSC

FTIR characterizations were carried out using an IRPrestige-21 spectrometer (Shimadzu, Kyoto, Japan) with attenuated total reflection (ATR) mode. The scanning range was varied from 500 to 4000 cm^−1^, and the resolution was 4 cm^−1^. Moreover, the OMNIC software was utilized for the data analysis.

Thermal properties and crystallinity of the XLPE specimens were measured by differential scanning calorimetry (DSC) instruments (Q200, TA Instruments Inc., New Castle, DE, USA) with a computer data system. Note that 10 mg of XLPE were analyzed in a temperature interval from 30 to 160 °C at a ramp rate of 10 °C/min after the elimination of thermal histories. The testing process is carried out in a nitrogen atmosphere.

### 2.3. VLF Dielectric Spectroscopy

The VLF dielectric responses of the XLPE samples from 1 mHz to 1 kHz were measured by the Insulation Diagnostic Analyzer (iFDS, Chongqing University, Chongqing, China), and the accuracy of iFDS is comparable with another commercial equipment such as Dirana (Omicron, Krause, Austria) [18,24]. The schematic diagram of this VLF test system is shown in Figure 2. The *U*–*I* hysteresis curves were obtained using the sweep mode of iFDS at a measuring voltage of 200 V, and the testing frequency of *U–I* hysteresis is set at 1 mHz and 1 kHz. A three-electrode test system was adopted in the dielectric test, in which the diameters of the high- and low-voltage electrodes were 60 mm and 48 mm, respectively. Moreover, a ring electrode was used as a shielding electrode to eliminate the interference from the leakage current along the sample surface.

## 3. Results and Discussions

### 3.1. Color Change of Samples

The photographs of XLPE samples at different aging stages are shown in Figure 3. It can be seen that there is no significant color change for samples in group A, while the color of samples in group B gradually changes from bright white to light yellow and become wrinkled as the aging time increased (120 °C—720 h). Moreover, a small number of brown patches appear on the edge of the samples in group C, while the brown area further develops toward the center of the sample with the development of aging (150 °C—480 h). Finally, the entire area of the sample (150 °C—720 h) turns dark brown with severe deformation. The change in color of XLPE insulation may be attributed to the generation of vinylidene groups, vinyl groups and other unsaturated groups in XLPE that are caused by thermal-oxidative aging [9,25].

### 3.2. FTIR Spectroscopy Analysis

FTIR is an effective technique to analyze chemical changes in the molecular chain structure of XLPE cables. For unaged XLPE, there are usually four characteristic bands in the FTIR spectrum with wavenumbers of 720 cm^−1^, 1460 cm^−1^, 2840 cm^−1^ and 2915 cm^−1^, corresponding to the rocking of methylene groups, wag vibration of methylene groups, stretching vibration of methyl groups and methylene groups, respectively [26].

The FTIR spectra of different thermal-aged XLPE samples are presented in Figure 4a–c. It can be seen that there was no significant change in the four characteristic bands after aging at 90 °C. As the aging temperature increased to 120 °C and 150 °C, the four bands decreased gradually, which could be a result of the scission of the molecular chain inside XLPE. It can be seen that the reflection peaks around 800 cm^−1^, 1020–1090 cm^−1^ and 1260 cm^−1^ obviously increase with aging degree [26]. Furthermore, the absorption peak at the wavenumber of 1720 cm^−1^ corresponding to carbonyl (-C=O) appears when the aging temperature reaches 120 °C and the intensity increases gradually with the thermal aging process, which means that the number of carbonyl groups increases [27,28]. The intensity of the absorption peak at 1720 cm^−1^ is strengthened obviously as the thermal temperature reaches 150 °C.

To quantify the degree of oxidation, the carbonyl index (*CI*) is calculated based on the following equation [12]:(1)$$CI=\frac{I_{1720}}{I_{2010}}$$
where *I*_1720_ and *I*_2010_ are the intensity of the absorption peaks at 1720 cm^−1^ and 2010 cm^−1^, respectively.

It can be seen from Figure 4d that the *CI* of the unaged cable insulation is about 0.37, which is caused by impurities during the manufacturing process of XLPE cables [29]. It is found that the aging at 90 °C had little effect on *CI*. As the aging temperature increases to 120 °C and 150 °C, a significant increase of *CI* was observed, indicating that the thermo-oxidation process occurs. In addition, it has been noticed that *CI* starts to increase rapidly after thermal aging at 120 °C for 240 h. The increase of *CI* indicates that the thermal aging process promotes the decomposition of XLPE microstructure and thus accelerates the generation of aging products [30].

### 3.3. DSC Results

DSC curves of the original and aged XLPE are shown in Figure 5a–c. For XLPE treated at 90 °C and 120 °C, the area and the intensity of the endothermic peak corresponding to the melting temperature of the crystalline part become narrow. During this aging process, the thermo-oxidation can be inhibited by the antioxidant. Since the oxidation mainly affects the amorphous regions, some lamellar crystals may melt, and their thickness was practically not affected. Moreover, some low-molecular-weight segments would appear in the amorphous region because of the scission of the ends of the macromolecular chains [9]. For the aging temperature at 150 °C, the endothermic peak of the XLPE sample shifts to lower temperatures (about 60 °C and 80 °C). At this stage, the antioxidant has been consumed, and thus the oxidation becomes auto-accelerated rapidly. The broad endothermic peaks can be attributed to the presence of smaller chain segments due to the main chain breakage [11,31].

From the results of Figure 5, the melting (*T_m_*) and the crystallization (*T_c_*) temperatures, the enthalpies of fusion ∆*H*_m_ and crystallization ∆*H*_c_ are obtained, as shown in Table 2, and the crystallinity rate (*X_c_*) is calculated using Equation (2),
(2)$$X_{c}=\frac{\Delta H_{m}}{\Delta H_{0}}\times 100\%$$
where ∆*H*_0_ = 287.3 J/g corresponds to the theoretical melting enthalpy of a complete crystallinity PE [32].

### 3.4. VLF Dielectric Spectroscopy

The VLF dielectric spectra of the thermally aged XLPE sample under different temperatures and various aging durations are shown in Figure 6 and Figure 7. The dielectric constant of all the samples remains steady in the frequency range of 0.1 Hz–1 kHz, while the values rise significantly in the lower frequency of 1 mHz–0.1 Hz. Moreover, the dielectric constant can be greatly affected by the thermal aging process. For example, the dielectric constant increases from 5 of the unaged sample to 23 of the aged sample (150 °C—720 h) at 1 mHz. As shown in Figure 7, tan*δ* of the XLPE samples with different aging times have a similar variation trend with frequency, i.e., tan*δ* shows a small difference at high frequency, but increases obviously at low frequency. For the XLPE samples aged at 90 °C, the tan*δ* begins to dramatically increase only when the test frequency is smaller than 0.02 Hz (characteristic frequency where the value of tan*δ* is 0.5). This significant increase of dielectric constant and tan*δ* is related to the carbonyl group produced by thermal aging.

After being aged at 150 °C, the characteristic frequency shifts to a higher frequency of 0.2 Hz, below which the tan*δ* is evidently enhanced. Therefore, the high tan*δ* value in the low-frequency range (1 mHz–0.1 Hz) can be defined as an indicator of XLPE insulation aging. It is believed that long-chain XLPE molecules break into short chains and thermal motion becomes intense during the aging process, which can result in more polar groups and carriers. These changes lead to more interfacial polarization (Maxwell–Wagner–Sillars), and make the dielectric relaxation phenomenon more obvious [33,34].

### 3.5. VLF Dielectric Spectroscopy of XLPE Samples

It is well known that dielectric loss is usually contributed from electric polarization and DC conductance. During the VLF region (*ωτ* << 1), the influence of leakage conductance on dielectric properties cannot be ignored, so the relationship between permittivity and test frequency follows:(3)$$\varepsilon^{*}(\omega )=\varepsilon^{\prime }(\omega )-j\varepsilon^{\prime \prime }(\omega )=\varepsilon_{\infty }+\frac{\varepsilon_{s}-\varepsilon_{\infty }}{1+\omega^{2}\tau^{2}}-j(\frac{\gamma }{\omega \varepsilon_{0}}+\frac{(\varepsilon_{s}-\varepsilon_{\infty })\omega \tau }{1+\omega^{2}\tau^{2}})$$
where *ω* is the angular frequency, *ε*′ and *ε*″ represent the real and imaginary part of permittivity, respectively. *ε_s_* and *ε*_∞_ are static and optical permittivity. *ε*_0_ is the vacuum permittivity, *γ* is the dc conductivity and *τ* is the relaxation time. However, when the insulating materials are seriously aged, the dc conductance will take a leading role in the dielectric loss at low frequency, which can be calculated as [35]:(4)$$\varepsilon^{*}(\omega )=\varepsilon^{\prime }(\omega )-j\varepsilon^{\prime \prime }(\omega )=\varepsilon_{s}-j\frac{\gamma }{\omega \varepsilon_{0}}$$

It can be found from Equation (4), that if the dc conductance flowing through the electrode interface obeys Ohm’s law, it indicates that the carriers do not accumulate at the electrode interface or the defect interface of the sample, and therefore no space charge interfacial polarization occurs. In this case, the contribution of the leakage conductance to the dielectric constant is constant and does not change with the frequency. At this point, the contribution of the conductivity to the dielectric loss is mainly in the low-frequency region, and the *ε*′ is inversely proportional to the frequency (*ω*^−1^). On the contrary, if the conductance flowing through the electrode interface does not follow Ohm’s law, it indicates that the carriers accumulate at the electrode interface or the defect interface of the sample, resulting in an additional interface polarization effect. The time constant of interface polarization is large, which usually occurs in the low-frequency region, resulting in a sharp increase of the dielectric constant. Therefore, considering the dc conductance and polarization, according to Equations (3) and (4), the tan*δ* can be described as
(5)$$\tan\delta =\frac{\varepsilon^{\prime \prime }(\omega )}{\varepsilon^{\prime }(\omega )}=\frac{(\varepsilon_{s}-\varepsilon_{\infty })\omega \tau }{\varepsilon_{s}+\varepsilon_{\infty }\omega^{2}\tau^{2}}+\frac{\gamma }{\varepsilon_{s}\varepsilon_{0}\omega }$$
where the first part corresponds to relaxation polarization, and the second part represents dc leakage conductance. According to the dielectric response of the severely aged sample C3 (thermal aging at 150 °C for 720 h), the FDS curve is fitted by Equations (3) and (5). The fitting result is shown in Figure 8, in which *ε*′_sum_ and tan*δ*_sum_ are the fitting data of cable insulation after thermal aging at 150 °C for 720 h, *ε*′*_s_* means the static permittivity, while *ε*′_interface_ represents the contribution of interface polarization which may be resulted from the space charge accumulation because of dc conduction between electrodes and sample, tan*δ*_interface_ represents the contribution of interface polarization and tan*δ*_dc_ is caused by the dc conduction process. It can be seen that the fitting results are in good agreement with the experimental data, and the dielectric constant is divided into two parts, the optical permittivity *ε*_∞_ and one “step” caused by the relaxation polarization. Similarly, tan*δ* is contributed by dc conduction and a relaxation peak.

### 3.6. Nonlinear U–I Hysteresis Curves

The nonlinear *U*–*I* feature is common in the dielectric response of aged XLPE cables. In this paper, the *U*–*I* hysteresis curve considering the relationship between a standard sinusoidal voltage and the current of different phases is introduced to evaluate the cable insulation [36]. The hysteresis curve is a standard ellipse for a new XLPE sample. However, when the sample is aged, the area, shape, and inclination of the major axis in the standard ellipse will change. Among them, the area and shape are related to the applied excitation voltage and the response current, showing the resistance of the XLPE samples. The change in inclination of the major axis is in connection with the phase difference between the excitation voltage and the response current.

The hysteresis curves of XLPE cable insulation slices are compared in Figure 9 under different thermal aging conditions at the frequencies of 1 kHz and 1 mHz. It can be observed that all hysteresis curves are approximately elliptical at the high frequency of 1 kHz. Compared to the unaged cables, the major axis of the hysteresis curve for thermally aged samples gradually shifts, and the current amplitudes at different phases increase simultaneously, especially for the samples in group C. As can be seen from Figure 9d–f, the hysteresis curves of the samples with different aging states show more obvious deformation in the low frequency (1 mHz). After being aged at 90 °C and 120 °C, the area and shape of the elliptic curve have no significant change, while the inclination of the major axis increases with the aging time. Moreover, the deflection of major axis and the shape of hysteresis curve change for the samples aged at a higher temperature. As shown in Figure 9f, the hysteresis curve area of the sample aged at 150 °C becomes larger with more serious deformation compared with the unaged cables at the same test frequency. These results indicate that the aging state of XLPE cables is strongly correlated to the deformation of the hysteresis curves measured at 1 mHz, which were attributed to the occurrence of higher harmonics in the response current at low frequencies [37].

## 4. Conclusions

This work studied the effect of the accelerated thermal aging on the physicochemical and dielectric properties of 10 kV XLPE cables. FTIR results show that the deformation and color change of the XLPE samples can be attributed to the thermo-oxidative reaction and accompanied with the generation of polar groups such as carbonyl groups. The crystallinity decreases, and the melting temperature shifts towards lower temperature after thermal aging of XLPE insulation. From the VLF results, it is found that the dielectric constant and dielectric loss are evidently enhanced in the low-frequency range below 0.1 Hz with the thermal aging process, which may be related to the increase of carbonyl. Moreover, the structural defects produced by thermal aging lead to an increase in interfacial relaxation, producing a significant relaxation peak at low frequencies. Finally, the *U–I* curve of the cable insulation after thermal aging shows the nonlinearity due to the high harmonics generated by the saturation of the response current. Therefore, the *U–I* hysteresis curve measured at 1 mHz can more effectively evaluate the insulation degradation of the XLPE cable.

## Figures and Tables

**Figure 1 materials-16-04388-f001:**
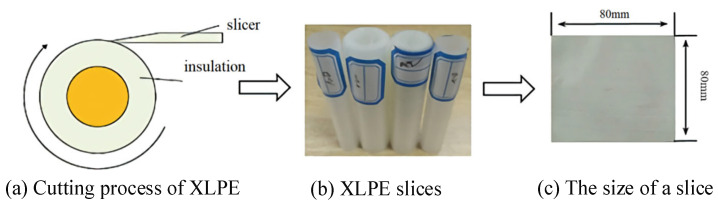
Preparation of XLPE slice samples.

**Figure 2 materials-16-04388-f002:**
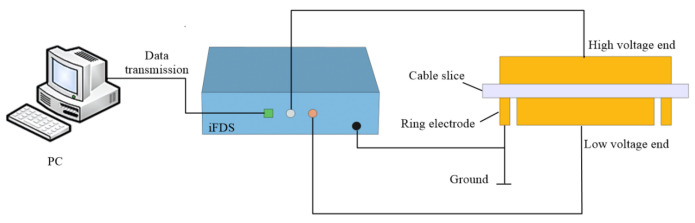
Schematic diagram of VLF dielectric spectroscopy for XLPE.

**Figure 3 materials-16-04388-f003:**
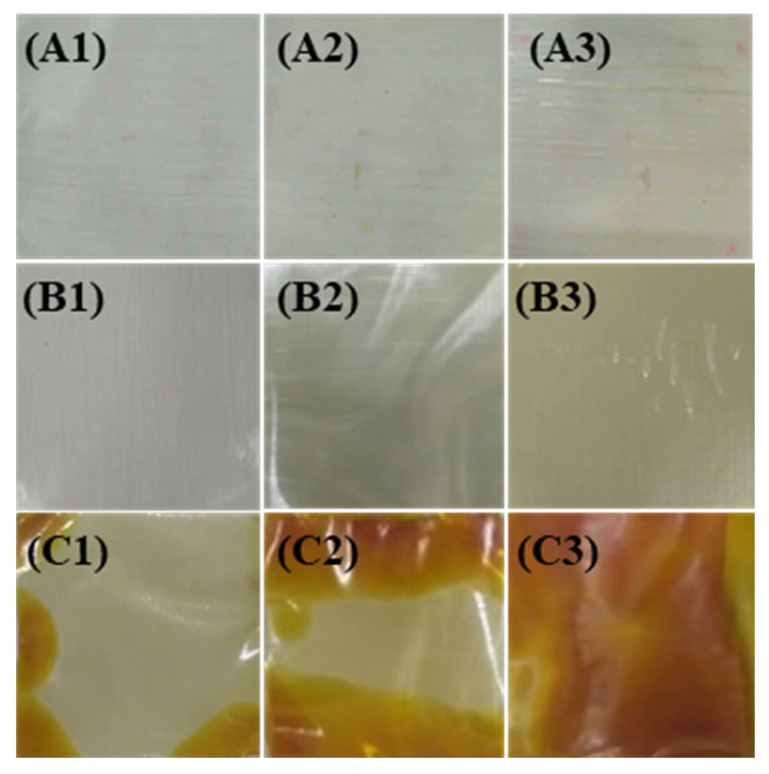
A1–C3 are aged at 90 °C, 120 °C and 150 °C for 240 h, 480 h and 720 h for XLPE cable slices, respectively.

**Figure 4 materials-16-04388-f004:**
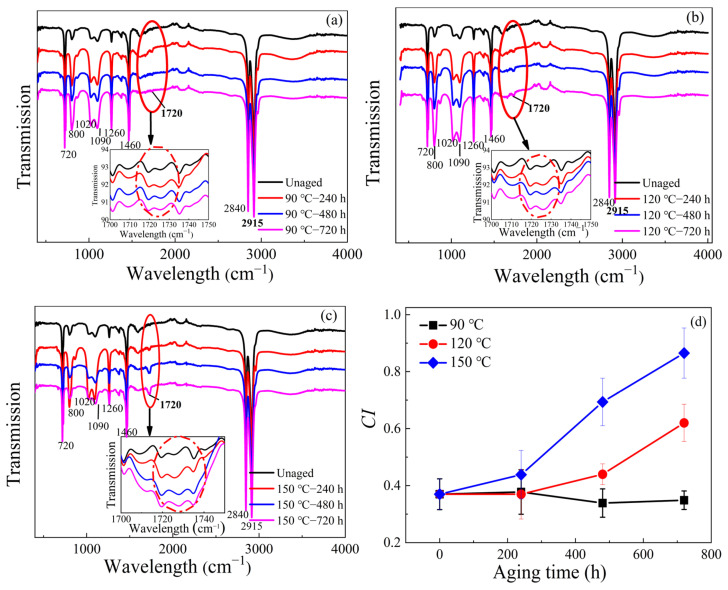
(**a**–**c**) FTIR spectra of XLPE samples with different thermal aging and (**d**) carbonyl index of XLEP samples with different thermal aging.

**Figure 5 materials-16-04388-f005:**
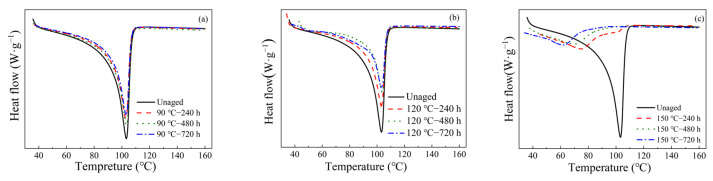
(**a**–**c**) DSC curves and the crystallinity of XLPE samples with different thermal aging levels.

**Figure 6 materials-16-04388-f006:**
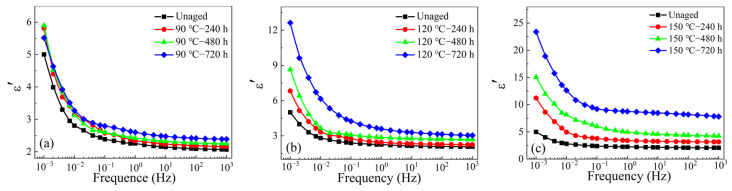
Effect of thermal aging on VLF dielectric constant of XLPE samples (**a**) aged at 90 °C, (**b**) aged at 120 °C, and (**c**) aged at 150 °C.

**Figure 7 materials-16-04388-f007:**
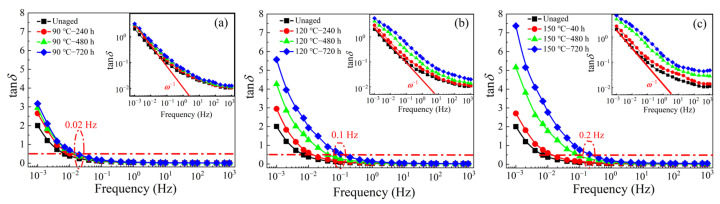
Effect of thermal aging on VLF dielectric loss (tan*δ*) of XLPE samples (**a**) aged at 90 °C, (**b**) aged at 120 °C, and (**c**) aged at 150 °C.

**Figure 8 materials-16-04388-f008:**
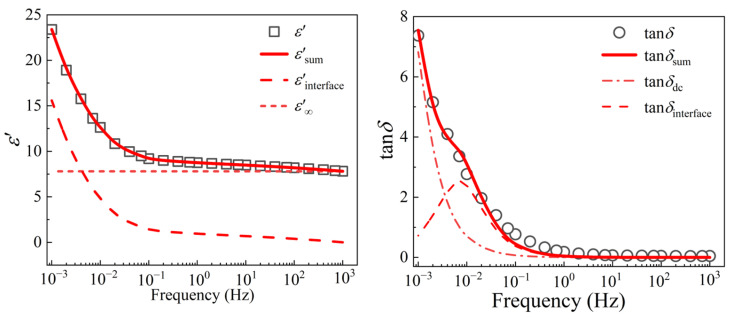
The fitting results of (**a**) *ε*′ and (**b**) tan*δ* of cable insulation after thermal aging for 720 h at 150 °C.

**Figure 9 materials-16-04388-f009:**
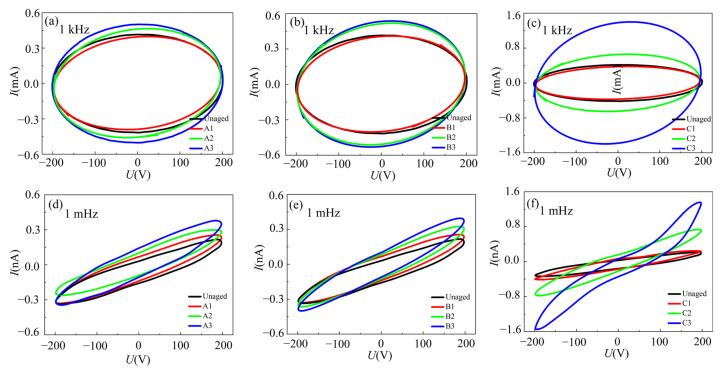
Hysteresis curves of XLPE cable slices with different frequencies. (**a**–**c**) the *U*–*I* hysteresis curve at 1 kHz of the aged sample of A1–C3. (**d**–**f**) the *U*–*I* hysteresis curve at 1 mHz of the aged sample of A1−C3.

**Table 1 materials-16-04388-t001:** Thermal aging parameters of XLPE samples.

Sample Number	Aging Temperature (°C)	Aging Time (h)
Unaged	/	/
A1	90	240
A2	90	480
A3	90	720
B1	120	240
B2	120	480
B3	120	720
C1	150	240
C2	150	480
C3	150	720

**Table 2 materials-16-04388-t002:** DSC parameters of XLPE samples with thermal aging.

Samples	*T_m_* (°C)	*T_c_* (°C)	∆*T_m_* (°C)	∆*H_m_* (J·g^−1^)	*X_c_*
Unaged	103.17	90.51	10.76	108.90	0.379
A1	103.27	91.02	10.37	103.00	0.359
A2	103.31	90.9	10.51	101.10	0.352
A3	103.15	91.1	10.11	102.47	0.357
B1	103.76	91.3	10.55	97.95	0.341
B2	103.77	92.28	8.14	76.75	0.267
B3	103.33	91.53	9.19	71.27	0.248
C1	75.39	57.3	45.31	57.96	0.202
C2	68.70	49.71	30.16	51.45	0.179
C3	62.23	43.7	27.56	42.60	0.148

## Data Availability

Not applicable.

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
