# Peer review of "Structural Changes and Very-Low-Frequency Nonlinear Dielectric Response of XLPE Cable Insulation under Thermal Aging"

_materials, 2023, doi:10.3390/ma16124388_

Round 1
Reviewer 1 Report
The paper discusses the structural changes and very-low-frequency nonlinear dielectric response of XLPE cable insulation under thermal aging. Nevertheless, the paper is not well written in its current form. The significance of the paper cannot be reflected through the current version of the paper. Some of my comments are as follows:
1. Please include the brand name and model of the air-circulating oven. Also, please include more details of the J/Q XLPE cable slicer used to cut the XLPE samples.
2. Please label and explain all the three pictures in Figure 1.
3. Please justify the reason behind the selection of 90 C, 120 C and 150 C aging temperatures and 240 h, 480 h and 720 h aging durations.
4. Please further explain the FTIR characterization modes. For the FTIR plot in Figure 4a-c, please refer to the standard way of plotting the wavenumber axis for the FTIR spectra.
5. The VLF dielectric responses were measured using the iFDS dielectric spectrometer made by Chongqing University, China. Is the equipment commercially available or custom made? This is because the paper revolves around VLF dielectric response and it is therefore crucial to provide detailed information about the equipment especially with respect to its VLF measurement capability.
6. Equations 3-5 and their discussion are theory. That should be introduced in the introduction section rather than in the discussion section.
7. Please provide more details about the data fitting for Figure 8. What do the authors mean by sum, interface and s symbols? Detailed explanation is required for Figure 8.
8. Please provide more details on how the hysteresis curves in Figure 9 were obtained. Detailed explanation is required for Figure 9.
English is understandable but can be improved.
Reviewer 2 Report
Comments:
1) Paragraph 2.2 FTIR and DSC: please provide more details abou: (a) the FTIR instrument (maker - it should be from "Shimadzu"? - not only the model) and the method used (transmission? DRIFT? ATR? - I presume ATR, but it need to be explicited clearly). What software was used for data analysis? (b) DSC: maker, model, etc. Software?
2) Paragraph 3.1 - color change. How the color change was evaluated? only visually? RGB values should be measured and compared, and the method appropriately described.
3) paragraph 3.2, line 120-121: only increase in the IR bands at 800, 1020-1090 and 1260 cm-1 are mentioned in the text; no mention of the band at 1720 cm-1 (which is used for carbonyl index determination). Please revise and correct this paragraph.
4) paragraph 3.3. DSC results needs revision and a more detailed discussion: it should be explained better that the melting peak is related to the non-crosslinked fraction (cross-linked fraction can not melt)
5) results should be discussed more extensively and conclusion paragraph could be expanded
Reviewer 3 Report
The paper introduces an investigation of the thermal degradation of XLPE cable insulation samples. The degradation processes were investigated by color change observation, FTIR, DSC, and VLF dielectric spectroscopy methods. The executed research work can be interesting, but improvement is necessary for the manuscript.
The introduction summarises some previous studies in the investigation of XLPE insulation. A short conclusion from the literature review needs to be included: Why has this research been done? What is the scientific gap that the study is intended to fill? Please improve this section.
Section 2 introduces the materials and methods. The sample preparation has been documented well, and the measurement equipment is clearly presented, enabling the reproducibility of the results.
Section 3 is about results and discussion. The results are presented clearly, and the graphs and tables are well-edited. The only weakness is that the charts in Figure 7 need to be bigger.
Unfortunately, the discussion of the results requires serious improvement. The results should have been evaluated in light of the existing literature. Please improve this section and explain the results of the study. In addition, in the section introduction, the authors wrote about the correlation between VLF tan delta and the thermal degradation of XLPE. I wonder why they did not test the correlation between the chemical tests (FTIR, CI, or DSC) and VLF tan delta to reject or confirm this hypothesis. Please insert a correlation analysis in this section. It would improve the quality and scientific soundness of the manuscript.
Based on the improvement of the discussion, please revise the conclusion, as well.
Minor corrections:
-Equation 3 cannot be found in the reference [25]
-Publication year of [26] is 1971, not 2003
Minor refinements are needed.
Round 2
Reviewer 1 Report
The authors have satisfactorily addressed most of my earlier comments. However, I hope the authors could further address the following comments that I made earlier:
1. Please add the justification for 240 h, 480 h and 720 h aging duration in the paper.
2. Please add the justification on the performance of the Insulation Diagnostic Analyzer (iFDS, Chongqing University, China), which is commercially available, in the paper.
3. The standard way of plotting the wavelength axis (x-axis) for the FTIR spectra is from the highest wavelength to the lowest wavelength. Please check. For the y-axis, I believe the authors mean "Transmission" and it is more appropriate to be without the % symbol since the number is not provided on the y-axis.
Reviewer 3 Report
Thank you for considering my suggestions. The paper has been improved, and I have no more questions or comments on the manuscript.
The English language is good. There are minor errors.
